Perioperative management of angiotensin-converting enzyme inhibitors and/or angiotensin receptor blockers: a survey of perioperative medicine practitioners

Walker Sophie L.M.
Abbott Tom E.F.
Brown Katherine
Pearse Rupert M.
Ackland Gareth L. g.ackland@qmul.ac.uk
William Harvey Research Institute, QMUL, Queen Mary University of London , London , United Kingdom
Walsh Stewart
Electronic publication date: 2018 Jun 29
Publication date: 2018
Volume: 6
Electronic Location ID: e5061
Received 2018 Feb 27; Accepted 2018 Jun 4
Copyright: ©2018 Walker et al.
Copyright year: 2018
Copyright holder: Walker et al.
License: This is an open access article distributed under the terms of the Creative Commons Attribution License, which permits unrestricted use, distribution, reproduction and adaptation in any medium and for any purpose provided that it is properly attributed. For attribution, the original author(s), title, publication source (PeerJ) and either DOI or URL of the article must be cited.
License URL: https://creativecommons.org/licenses/by/4.0/

Keywords: Anaesthesia, Myocardial injury, Hypotension, Noncardiac surgery, Angiotensin-II, Survey, Anesthesiology

Funding: NIHR British Oxygen Company research chair in Anaesthesia Medical Research Council and British Journal of Anaesthesia R/M017974/1 Sophie L.M. Walker is supported by a NIHR research studentship. Gareth L. Ackland is supported by the British Oxygen Company research chair in Anaesthesia. Rupert M. Pearse is supported by a NIHR research professorship. Tom E.F. Abbott is supported by a Medical Research Council and British Journal of Anaesthesia clinical research training fellowship (grant reference R/M017974/1). The funders had no role in study design, data collection and analysis, decision to publish, or preparation of the manuscript.

==============================
Background

Angiotensin-converting enzyme inhibitors (ACEi) and angiotensin receptor blockers (ARB) are the most commonly prescribed antihypertensive medications in higher-risk surgical patients. However, there is no clinical consensus on their use in the perioperative period, in part, due to an inconsistent evidence-base. To help inform the design of a large multi-centre randomized controlled trial (ISRCTN17251494), we undertook a questionnaire-based survey exploring variability in ACEi/ARB prescribing in perioperative practice.

Methods

The online survey included perioperative scenarios to examine how consistent respondents were with their stated routine preoperative practice. Clinicians with an academic interest in perioperative medicine were primarily targeted between July and September 2017. STROBE guidelines for observational research and ANZCA Trials Group Survey Reporting recommendations were adhered to.

Results

194 responses were received, primarily from clinicians practicing in the UK. A similar minority of respondents continue ACEi (n = 57; 30%) and ARBs (n = 62; 32%) throughout the perioperative period. However, timing of preoperative cessation was highly variable, and rarely influenced by the pharmacokinetics of individual ACE-i/ARBs. Respondents’ stated routine practice was frequently misaligned with their management of common pre- and postoperative scenarios involving continuation or restarting ACE-i/ARBs.

Discussion

This survey highlights many inconsistencies amongst clinicians’ practice in perioperative ACE-i/ARB management. Studies designed to reveal an enhanced understanding of perioperative mechanisms at play, coupled with randomised controlled trials, are required to rationally inform the clinical management of ACE-i/ARBs in patients most at risk of postoperative morbidity.

Introduction

Over 1.5 million high-risk patients undergo surgery in the UK every year (Abbott et al., 2017a). Angiotensin-converting enzyme inhibitors (ACEi) and angiotensin receptor blockers (ARB) are the most commonly prescribed antihypertensive medications in high-risk surgical patients (Abbott et al., 2017b; Ackland et al., 2015; Pearse et al., 2014). While the benefits of ACEi/ARBs in hypertension, ischaemic heart disease, heart failure, diabetes mellitus and renal disease are well-recognised (SPRINT Research Group et al., 2015; Heart Outcomes Prevention Evaluation Study Investigators et al, 2000; Yusuf et al., 2003), continuation of their use throughout the entire perioperative period remains controversial.

Directly conflicting data raise significant doubts about the perioperative management of ACEi/ARBs in noncardiac surgery. Some, but not all, early-phase observational clinical studies (Brabant et al., 1999; Schulte et al., 2011) report association between ACEi/ARBs and perioperative hypotension, according to a variety of definitions (Abbott et al., 2017c; Bijker et al., 2007; Vaquero Roncero et al., 2017). However, these data contrast with the results of large clinical database studies using objective outcome measures, which provide markedly divergent findings in noncardiac surgery. Both failure to restart ACEi/ARB therapy after surgery (Lee, Takemoto & Wallace, 2015; Mudumbai et al., 2014), and not stopping ACEi/ARBs before surgery (Roshanov et al., 2017), have been associated with increased incidence of postoperative mortality in both noncardiac and cardiac surgery. Conversely, the Cleveland Clinic Outcomes Research group found no association between perioperative ACEi use and the incidence of postoperative complications or mortality in >79,000 patients undergoing non-cardiac surgery (Turan et al., 2012). In keeping with these conflicting data, three systematic reviews conclude that the evidence surrounding perioperative ACEi/ARB use is characterised by retrospective, observational studies of low methodological quality, high risk of bias and a lack of power to explore objective outcomes (most notably, postoperative morbidity) (Hollmann, Fernandes & Biccard, 2018; Vaquero Roncero et al., 2017; Zou et al., 2016). Moreover, the failure to take into account the variable pharmacokinetic characteristics of ACEi/ARBs may be an additional confounder (Michel et al., 2013).

To help inform the design and conduct of a randomised controlled trial now underway (ISRCTN17251494), we undertook a survey to determine the current state of practice and opinions regarding perioperative ACEi/ARB use in noncardiac surgery. Our data indicate that there is widespread uncertainty regarding the perioperative use of ACEi/ARB. This reinforces the need for a randomized controlled trial to inform clinical practice.

Methods

Study design

We used the PICO framework to design this online survey of practice regarding the perioperative use of ACEi/ARB, as detailed in Table 1. The study received research ethics approval (QMREC1735; Queen Mary, University of London Ethics of Research Committee) and was conducted in accordance with the principles of the Declaration of Helsinki and the Research Governance Framework. We adhered to STROBE guidelines for observational research, although these do not include reporting characteristics that are specific for surveys (Von Elm et al., 2007). In addition, we also used the ANZCA Trials Group Survey Reporting List (Appendix S1) to ensure transparency and reproducibility (Story et al., 2011).

Table 1 PICO model to frame questions for survey.

Patient:	Patients undergoing non-cardiac surgery, requiring ACE-inhibitor and/or angiotensin receptor blocking therapy for cardiometabolic and/or renal disease.	
Intervention:	Withdrawal of ACE-inhibitor and/or angiotensin receptor blockade therapy.	
Comparison:	Continuation of ACE-inhibitor and/or angiotensin receptor blockade therapy.	
Outcome:	Identify perioperative practice under different common clinical scenarios.	

Questionnaire design

The survey consisted of 11 questions designed to ascertain opinions regarding perioperative ACEi/ARB use and illicit routine perioperative prescribing patterns. Questions were structured using two complementary approaches (Appendix S2). Firstly, five questions had a constructed response format, which were designed to directly address perioperative prescribing patterns. Secondly, six questions were based on common perioperative scenarios, which aimed to examine whether simple clinical guidelines for these drugs are easily generalisable. This section entailed several common clinical challenges including pre- and postoperative hypertension, early perioperative myocardial injury and relative postoperative hypotension (see Appendix S2). The final three questions addressed respondents, current grade and location of practice as well as their clinical background. The survey was constructed by a writing group (KB, TA, GA) and was initially piloted amongst anaesthetists at The Royal London Hospital, Barts Health NHS Trust. The survey underwent external peer review and second-round piloting at the Health Services Research Centre, Royal College of Anaesthetists, UK. The final version of the survey was approved for clarity and feasibility by all authors.

Participants and survey administration

Participation in the survey was voluntary and responses were anonymised. Participants were identified and invited via three routes: 33 attendees at the Perioperative Quality Initiative (POQI) Consensus Conference on perioperative management of arterial blood pressure (2017), 161 principal investigators from 120 UK centres that contributed to the International Surgical Outcomes Study (ISOS; Appendix S3) and an open invitation to members of the Royal College of Anaesthetists (issued on 26 June 2017; https://www.rcoa.ac.uk/rcoa-presidents-news-june-2017). Detailed information on the study was provided in a letter sent by e-mail explaining the goals and design of the study, specifying confidentiality and the handling of data. Formal written consent was not required before participation. However, consent was implied through participation. Participants answered the questions via an online survey (SurveyMonkey). Two e-mail reminders were sent to optimise the response rate.

Statistical methods

Categorical data are presented as n (%), analyzed by Fishers exact test, and presented as odds ratios (95% confidence intervals). All reported p values are two-sided, with significance defined by p values ≤0.05. Statistical analyses were performed using GraphPad Prism software (La Jolla, CA, USA).

Results

194 anonymised surveys were collected in total between 26 June 2017 and 31 September 2017, with a 64.4% response rate from 125/194 POQI/ISOS investigators (Fig. S1). 183 (96%) respondents practiced perioperative medicine within the United Kingdom and 163 (84.5%) were consultants in perioperative medicine/anaesthesia.

Planned preoperative prescription

A similar minority of respondents continue ACEi (n = 57; 30%) and ARBs (n = 62; 32%) throughout the perioperative period. Amongst the 135 (70%) participants who routinely stop ACEi prior to surgery, 97 (72%) routinely recommend ACEi/ARB cessation on the day of surgery. Only three (2%) participants base ACEi/ARB cessation on the half-life of the drug (Fig. 1).

Figure 1 Routine pre-operative practice for ACEi and ARB.

Planned postoperative prescription

Correspondents reported that a variable time at which ACEi/ARBs were restarted after surgery (Fig. 2). 76 (40%) and 78 (41%) respondents would restart ACEi or ARBs within 24 h of major surgery, respectively. The practice of routinely stopping ACEi/ARBs was associated with >24 h delay in restarting ACEi/ARBs (odds ratio: 3.44 (95% CI [1.81–6.41]); p < 0.001; Fig. 3).

Figure 2 Routine post-operative practice for ACEi and ARB use.

Figure 3 Association between planned pre- and post-operative practice.

Routinely stopping ACEi/ARBs in the pre-operative setting was associated with a > 24 hr delay in restarting the drug (odds ratio: 3.44 (95% CI [1.81–6.41]); p < 0.001).

Management of preoperative hypertension, on the day of surgery (questions 6, 7, 8)

When faced with the scenario of an acutely hypertensive patient (mean arterial pressure >160 mmHg) immediately before surgery, respondents did not appear to consider the preoperative use of ACEi/ARB as an important influence on proceeding to surgery. For patients normally on ACEi/ARB who had stopped the drug pre-operatively, there was no association between respondents’ usual pre-operative practice (continuing versus stopping ACEi/ARB) and decision to proceed to surgery on that day (OR:2.37 (95% CI [0.85–6.2]); p = 0.10; question 7; Fig. 4). Untreated hypertension was more likely to trigger postponement of surgery than hypertension in patients already prescribed ACEi/ARB (OR:1.71 (95% CI [1.07–2.73]); p = 0.03; question 6; Fig. 4).

Figure 4 Clinical scenarios: survey questions 6–9.

Management of postoperative blood pressure in the early postoperative period (question 9)

169 (87%) respondents declined to restart ACEi/ARBs within 24 h in a stable, high-risk patient on chronic ACEi/ARBs if their systolic blood pressure was 90–100 mmHg. For respondents who advocated continuing ACEi/ARBs throughout the perioperative period, faced with this scenario only 11 (20%) would continue ACEi/ARBs. However, these respondents who advocated continuing ACEi/ARBs were three times more likely to restart the drug (OR:3.17 (95% CI [1.26–8.24]); p < 0.05; Fig. 4).

Management of suspected postoperative myocardial injury (questions 10, 11)

Postoperative hypertension (systolic arterial pressure >170 mmHg) associated with clinically asymptomatic rise in plasma high-sensitivity troponin on postoperative day one prompted 110 (57%) of respondents to restart ACEi/ARB in patients established on this therapy (Fig. 5). By contrast, less than 6% commenced ACEi/ARB in patients with a similar postoperative picture who were not already receiving ACEi/ARBs, deferring to specialist advice.

Figure 5 Impact of postoperative myocardial injury on ACEi/ARB use.

An asymptomatic rise in troponin post-operatively would prompt 110 (57%) to restart ACEi/ARBs in those already established on this therapy. In those patients not already receiving ACEi/ARBs, a similar asymptomatic troponin rise prompts only 6% to commence the drug deferring to specialist advice.

Discussion

The principal finding of this study is apparent widespread uncertainty surrounding the perioperative management of ACEi/ARB, even amongst experienced clinicians with an academic interest in perioperative blood pressure control and postoperative outcomes. This may be partly due to the different indications for ACEi/ARB therapy. This mirrors the conclusions of three independent systematic reviews that were unable to provide any recommendation on perioperative management of ACEi/ARB, chiefly due to poor study design and the lack of objective outcomes (Hollmann, Fernandes & Biccard, 2018; Vaquero Roncero et al., 2017; Zou et al., 2016).

Surgical patients with cardiac failure are at high-risk of postoperative morbidity and mortality (Abbott et al., 2016; Abbott et al., 2017c; Abbott et al., 2017d; Hammill et al., 2008; Hernandez et al., 2004). Moreover, many surgical patients are deconditioned and share strikingly similar cardiopulmonary physiology with cardiac failure patients (Abbott et al., 2017b). As revealed by cardiopulmonary exercise testing, many of these surgical patients have impaired left ventricular function- even though they have no formal diagnosis of cardiac failure. Registry data for hospitalized, cardiac failure patients show that 30-day mortality was substantially higher in those in whom ACEi/ARBs were discontinued (adjusted hazard ratio [HRadj] 1.92; 95% CI [1.32–2.81]; P < 0.001), with the readmission rate post-discharge lowest among patients continued or started on therapy (Gilstrap et al., 2017). These marked outcome differences persisted after discharge, with higher one-year mortality (41.6%) associated with discontinuation of ACEi/ARBs (HRadj 1.35; 95% CI [1.13–1.61]; P < 0.001).

Our survey also highlighted that individualisation of preoperative ACEi/ARB management is seldom considered by clinicians. ARBs have very different terminal half-lives, receptor binding kinetics, active metabolic components, highly variable volumes of distribution and some exhibit insurmountable antagonism, compared to ACEi (Michel et al., 2013). Thus, a one-size-fits-all approach for perioperative ACEi/ARB management is illogical. However, studies in which perioperative ACEi/ARB therapy was stopped have failed to consider the type of ACEI/ARB, patient indication for therapy or the time point at which ACEi/ARB therapy was restarted.

Strengths of this survey, which is the largest undertaken thus far, include deliberately targeting experienced clinicians (ISOS/POQI investigators) with an academic interest in perioperative blood pressure control and postoperative outcomes. A further strength was that, regardless of clinical/academic background, we probed the internal consistency of each respondents, pre- and postoperative practice by further exploring specific scenarios. This approach frequently revealed a disconnect between respondents, stated routine practice and their answers to common postoperative scenarios. The survey is limited by a relatively small sample that may not be representative of global practice, which is heavily geographically biased towards the UK. The survey would be strengthened by corroborative clinical data to assess whether respondents managed patients in a real world setting in a similar manner to that reported in the survey. In accordance with ethical committee requirements for survey responses to be completely anonymised, we cannot verify the exact origin of correspondents. It is likely that more detailed scenarios including renal function may have revealed more nuanced responses.

In conclusion, this survey suggests that the clinical management of ACEi/ARB therapy is highly variable, and often internally inconsistent. This survey, and systematic reviews, highlight the need for a mechanistic randomized controlled trial, using blinded outcomes in patients most at risk of postoperative morbidity.

Supplemental Information

Appendix S1 Survey checklist and respondence rate

Click here for additional data file.

Appendix S2 Survey monkey questionnaire

Click here for additional data file.

Appendix S3 UK ISOS investigator centres

Each centre had one potential correspondent principal investigator, unless stated otherwise.

Click here for additional data file.

Data S1 Survey results- raw data

Click here for additional data file.

Figure S1 Flow diagram showing eligible correspondents and response rate

Click here for additional data file.

Additional Information and Declarations

Competing Interests

Author Contributions

Ethics

Data Availability

Gareth L. Ackland is an Editor for the British Journal of Anaesthesia. Rupert M. Pearse has given lectures and/or performed consultancy work for GSK, Nestle Health Sciences, BBraun, Medtronic and Edwards Lifesciences. Rupert M. Pearse is a member of the Associate Editorial Board of the British Journal of Anaesthesia.

Sophie L.M. Walker conceived and designed the experiments, performed the experiments, analyzed the data, prepared figures and/or tables, authored or reviewed drafts of the paper, approved the final draft.

Tom E.F. Abbott conceived and designed the experiments, performed the experiments, analyzed the data, authored or reviewed drafts of the paper, approved the final draft.

Katherine Brown performed the experiments.

Rupert M. Pearse contributed reagents/materials/analysis tools, authored or reviewed drafts of the paper, approved the final draft.

Gareth L. Ackland conceived and designed the experiments, performed the experiments, analyzed the data, contributed reagents/materials/analysis tools, prepared figures and/or tables, authored or reviewed drafts of the paper, approved the final draft.

The following information was supplied relating to ethical approvals (i.e., approving body and any reference numbers):

The study received research ethics approval (QMREC1735; Queen Mary, University of London Ethics of Research Committee) and was conducted in accordance with the principles of the Declaration of Helsinki and the Research Governance Framework.

The following information was supplied regarding data availability:

The raw data are provided in a Supplemental File.

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
