# Peer review of "Perioperative management of angiotensin-converting enzyme inhibitors and/or angiotensin receptor blockers: a survey of perioperative medicine practitioners"

_PeerJ, doi:10.7717/peerj.5061_

## Round 0.1 · original submission · Minor Revisions

Thank you for submitting this interesting and useful article. The reviewers have made a number of suggestions. In particular, the issues relating to internal consistency raised by Reviewer 2 should be addressed in some detail in the discussion. As both reviewers mention, some further information on the survey respondents would be helpful for the reader.

Reviewer 1 ·

Basic reporting

clear and umabigous, professional English used throughout.

Experimental design

design of the survey is very good.

Validity of the findings

restricted to limited area (UK only) and based on relatively small number of responders (however acknowledged in limitation section) - and so far the largest available

Additional comments

Review: „Perioperative management of angiotensin-converting
enzyme inhibitors and/or angiotensin receptor blockers: A survey of specialists in perioperative medicine”

Interesting survey study in an important topic for the perioperative medicine.
I have some comments that I would like the authors to respond:

1. Several metaanalyses have been performed in the withholding/continuing ACEI/ARB. Many of them cited by the authors. There is one, most recent in Aensthesia Analg 2018 by Hollmann et. al, that the authors may consider to add to their thorough background introduction – as it is the most recent.
2. Abstract and Conclusions – it is maybe too much to say that ‘these data reinforce recent meta-analyses that support..... need for a mechanistic blinded randomized controlled trial’. I would be cautious getting these conclusions based on such a small sample sized survey limited to one geographical region (acknowledged by the authors in the limitation section).
3. Results
- It would be interesting to see the exact numbers of participants from each group (i.e POQI conference, PIs of ISOS, and open invitation) who responded to the invitation to fill the survey
- As the results for ACEI and ARB are almost identical, the authors may consider to show them together, and be consistent in it throughout their paper (example fig 1 with division whereas figure 2 without)
- Probably there are too many figures in this study. Some of them do not make much sense to the Reviewer. I am not sure it is of an importance to show the comparison of restarting ACEi/ARB between the groups of responders that stop vs do not stop them before surgery. Those that do not stop them – do not have to restart them? Perhaps enough to show when those who stop them restart them after the surgery (no need for statistical comparison)

·

Basic reporting

1. References: good reference for differences in actions of ACE-I and ARBs (see text below)
2. Flow diagram of participants would be desirable.
3. Please split figures 1a and 1b into figures 1 and 2 (see reasons in text below)
4. A PICO question is needed to frame the survey question (see text below)

Experimental design

1. Within the scope of the journal
2. The PICO for the survey should be added (in text below)
3. The internal consistency planned analyses should be added to statistical methods (in text below):

Validity of the findings

1. I am not convinced that the Q6-11 adequately determine consistency with Q1-5 (please see text below).
2. Q6-11 do show that the decision to proceed/ manage ACE-I/ARBs is complicated and not easily generalisable to simple clinical guidelines.

Additional comments

Dear Colleagues

Thank you for the opportunity to review the paper; ‘Perioperative management of angiotensin-converting enzyme inhibitors and/or angiotensin receptor blockers: A survey of specialists in perioperative medicine’. This is an electronic survey of the view of physicians with an interest in perioperative medicine, and their management of ACE-I and ARBs in the perioperative period.

The objective of this paper is to establish the current practice of perioperative medicine physicians and then to determine if the actions in a clinical scenario are consistent with declared practice. The findings of this study suggest that the proposed standard clinical practice, and the actual practice of the perioperative physicians shows little correlation i.e. little internal consistency.

A major limitation of the paper however, is that the survey respondent options available to determine i) normal practice, and ii) the scenarios to determine internal consistency, were not the same. Therefore, in normal practice questions (Q1-5), the questions implied that the patient was going for surgery (no option for postponing), but in the clinical scenarios the option to postpone surgery was allowed. It appears therefore that the difference between the two, may be more pronounced (significant) because of the structure of the possible responses allowed i.e. it would have been less different if ‘postponing’ surgery had been an acceptable response in Q1-5.

I do however, believe that this survey illustrates that the clinical decision whether to withhold, or administer ACE-I/ARBs is very dependent on the clinical scenario that the physician faces, as opposed to a standard perioperative regimen.

Specific comments:
1. Title: I don’t believe one can state; ‘specialists in perioperative medicine’, as the majority of respondents are not accredited ‘specialists in perioperative medicine’, and not all respondents were ‘specialists/ consultants’. Maybe; ‘perioperative medicine practitioners’
2. Abstract:
a. Line 33: Please put the denominator for the responses.
3. Introduction:
a. Line 50: Editorial manager has given the author name as ‘Group’.
b. Line 52: You need to address the type of surgery; i.e. non-cardiac, or all surgeries.
c. Line 72: Please add a PICO (population, intervention, comparison, outcome) question for the survey. It would clarify for the reader the objective of the survey, and clarify the surgical focus of the survey. (I see that this is clarified in the survey as major noncardiac surgery; https://www.rcoa.ac.uk/rcoa-presidents-news-june-2017.)
4. Methods:
a. Line 90: Please add, that it included three questions about the respondents.
b. Line 109: What is the membership of the Royal College of Anaesthetists? Please state the number of members that the open invitation went to.
c. Statistical methods, Line 117 onwards: Please define what were the pre-planned internal consistency analyses were in the study.
5. Results:
a. Line 125: The open invitation to the RCA is not adequately addressed. I realise it makes it difficult to determine the real denominator, as there is overlap between the 3 groups, and one has no idea of the number of RCA members actually read the newsletter. But we do need a better idea of where the 194 responses came from. A flow diagram with all three groups canvassed would be desirable.
b. Line 129 and 136: I would recommend putting the question number in brackets in the title, as you have done in line 143.
c. Line 143: My main concern with these questions as tests of internal consistency is that;
i. They now include a description of a patient with a low AT in a patient for elective surgery i.e. heart failure in addition hypertension. The scenarios in Q1-5 were essential focussed on blood pressure, and not heart failure.
ii. Now, the respondent can also postpone the surgery. In Q1-5, the respondent could not.
iii. As a result, I am not convinced that these responses actually determine internal consistency with Q1-5. Rather I believe that they demonstrate that clinical practice is far more complicated when determining practice for an individual patient, than general principles of how to manage a drug in the perioperative period (i.e. a simple practice guideline).
d. Line 150: I would contest that untreated heart failure (low AT) was possibly also a strong driver to postponing surgery, than untreated hypertension alone.
e. Line 165: This suggests that ‘perioperative physicians’ are uncomfortable at initiating these drugs. This is an interesting finding.
6. Discussion:
a. Line 176: I would suggest leading into this discussion, that there is widespread uncertainty regarding the management of ACE-I and ARBs, and that this may be partly due to the different indications for the therapies.
b. Line 188: Again, I believe this is also partly due to the different indications for the therapies.
c. Line 191: A good reference for this sentence would be good.
7. Figures
a. I would suggest splitting figure 1a and 1b into figures 1 and 2, as figure 1b does not represent the entirely identical cohort i.e. there are no patients who have continued their therapy in figure 1b.
8. Supplementary files
a. For the questionnaire, I would suggest adding the preamble, that one can see if you click on the survey monkey link.

Thank you for the opportunity to review this study.

Regards
Bruce Biccard

---

## Round 0.2 · Minor Revisions

The manuscript is much improved but please address the remaining minor points raised by Reviewer 2

Reviewer 1 ·

Basic reporting

clear

Experimental design

original research suitable for the journal

Validity of the findings

conclusions well stated, rather new information.

Additional comments

the authors have addressed all my comments.

·

Basic reporting

Acceptable

Experimental design

Acceptable

Validity of the findings

Acceptable

Additional comments

Dear Colleagues

Thank you for the opportunity to review a revision of the paper; ‘Perioperative management of angiotensin-converting enzyme inhibitors and/or angiotensin receptor blockers: A survey of specialists in perioperative medicine’. This is an electronic survey of the view of physicians with an interest in perioperative medicine, and their management of ACE-I and ARBs in the perioperative period.

The objective of this paper is to establish the current practice of perioperative medicine physicians and then to determine if the actions in a clinical scenario are consistent with declared practice. The findings of this study suggest that the proposed standard clinical practice, and the actual practice of the perioperative physicians shows little correlation i.e. little internal consistency.

I believe that the authors have largely addressed my concerns raised in the original submission. I have some very minor comments on this revision.

Minor comments:
1. Introduction:
a. Line 45: Editorial manager still gives the author name as ‘Group’.
2. Results:
a. Line 120: Please change ‘Supplementary data’ to ‘Supplementary Figure 1’
b. Management of preoperative hypertension, on the day of surgery (questions 6, 7, 8): As you transpose the results of questions 6 (line 145) and 7 (line 143) in the reporting of the odds ratios (i.e. question 7 is presented before question 6), I would suggest highlighting which question each odds ratio refers to.
3. Figures
a. I am not sure what the titles are for the figures, as they merely reproduce the text from the results. A simple title/ legend for each figure is needed.
4. Supplementary files
a. When referring to ‘supplementary data’ in the manuscript, it would be preferable if you were specific e.g. Supplementary Figure 1.

Thank you for the opportunity to review this revision.

Regards
Bruce Biccard

---

## Round 0.3 · accepted · Accept

Thank you for making these revisions.